# Larval Crowding Did Not Enhance Adult Migration Propensity in *Spodoptera frugiperda*

**DOI:** 10.3390/insects13070581

**Published:** 2022-06-26

**Authors:** Weixiang Lü, Linghe Meng, Xingfu Jiang, Yunxia Cheng, Lei Zhang

**Affiliations:** 1State Key Laboratory for Biology of Plant Diseases and Insect Pests, Institute of Plant Protection, Chinese Academy of Agricultural Sciences, Beijing 100193, China; lvwx@cwnu.edu.cn (W.L.); cranemeng@126.com (L.M.); xfjiang@ippcaas.cn (X.J.); yxcheng@ippcaas.cn (Y.C.); 2Key Laboratory of Southwest China Wildlife Resources Conservation, China West Normal University, Nanchong 637002, China

**Keywords:** *Spodoptera frugiperda*, larval crowding, flight capacity, reproduction, cannibalism

## Abstract

**Simple Summary:**

The fall armyworm, *Spodoptera frugiperda* (J. E. Smith), is a typical invasive migratory pest with a strong reproductive capacity, which has caused serious damage to crops. Larvae of *S. frugiperda* exhibit high levels of cannibalism associated with larval crowding. However, the response of *S. frugiperda* adults to such stress remains unclear. In this study, we investigated the effects of larval density on flight and reproductive parameters of *S. frugiperda* adults. We found that *S. frugiperda* reared under high-larval density conditions exhibited lower pupal and body weights, weaker flight and reproductive capacities than those reared as solitary larvae. This result was different from that of many migratory insects, where larval crowding enhanced migratory propensity of subsequent adults. In contrast, high-larval density conditions did not increase adult migration propensity in *S. frugiperda*. These findings enhance our understanding of migratory and reproductive behaviors of *S**. frugiperda* in response to larval density.

**Abstract:**

Reproduction and flight are two major adaptive strategies to cope with environmental stress in migratory insects. However, research on density-mediated flight and reproduction in the global migratory agricultural pest *Spodoptera frugiperda* is lacking. In this study, flight and reproductive performances in response to larval crowding were investigated in *S. frugiperda*. We found that larval crowding significantly reduced the pupal and body weights of *S. frugiperda*. Adults reared under the highest density of 30 larvae/jar had the minimum wing expansion, which was significantly smaller than that of larvae reared under solitary conditions. Larval crowding also significantly increased the pre-oviposition period (POP) and period of first oviposition (PFO) but decreased the lifetime fecundity, flight duration and flight distance. Our results showed that *S. frugiperda* reared under solitary conditions exhibited higher pupal and body weights and stronger reproductive and flight capacities than those reared under high-density conditions. Larval crowding did not enhance the migration propensity in *S. frugiperda* adults. In conclusion, larval crowding may not be a major factor affecting FAW migration due to high levels of cannibalism. These findings provide new insights into the population dynamics of *S. frugiperda* under larval crowding conditions.

## 1. Introduction

Larval density is one of the important abiotic factors shaping the behavior and life history of organisms. In holometabolous insects, larval crowding increases resource competition and the risk of cannibalism (namely, the killing and consumption of conspecifics) and can thus affect individual fitness-related traits [1,2,3], such as growth, reproduction, flight and adult longevity [4,5,6,7]. Many migratory insects, such as gregarious or solitary locusts [8] and short-winged or wingless crickets [9], have evolved from the morphs of migrants and residents in response to larval crowding, starvation and other stressful conditions [10,11,12].

Migration is usually considered an important life-history trait to cope with adverse environmental conditions [13]. Indeed, evidence of migration and reproductive traits associated with larval density has been reported in previous studies [10,14,15]. Individuals with high larval density experience tend to develop into migrants that have poorer reproductive performance and stronger flight capacity than residents [16], such as *Mythimna separata* (Walker) [17,18] and *Loxostege sticticalis* L. [10]. However, several authors reported that larval crowding decreased the flight performance of adults in *Agrotis ipsilon* (Hufnagel) [19] and *Cydia molesta* (Busck) [20]. Many studies have shown that larval crowding reduces the lifetime fitness of some migratory moths by decreasing the survival rate, body mass and fecundity [21,22]. In addition, higher population density regarded as a trigger of cannibalistic behavior has been reported in half of herbivorous insects, including *Drosophila melanogaster* (Guenee) and *Tribolium castaneum* (Herbst) [23,24]. Cannibals can gain direct benefits that include reduced competition and added acquisition of rich nutrients, which is more conducive to improving their own survival and reproduction [3,25]. In *Mallada basalis* (Walker), cannibalism can enhance the development and fecundity [26]. However, it may also increase the risk of pathogen and parasite transmission between populations [27]. In some lepidopteran insects, larvae living in nutrient-rich conditions exhibit high levels of cannibalism, resulting in the decline of fitness, including prolonged larval duration and decreased pupal weight [28], such as *Spodoptera exigua* (Hübner) [29] and *Helicoverpa*
*armigera* (Hübner) [30].

The fall armyworm (FAW, *Spodoptera frugiperda* J. E. Smith) is a typical, migratory and destructive agricultural pest with a world-wide distribution. Due to its long-distance migration, high dispersion rate, generalist food habits and strong reproductive capacity, the FAW poses a serious threat to global food security [31,32,33]. FAW first invaded Africa in 2016 [34,35], India and Southeast Asia in 2018 [36,37], and China in 2019 [38,39]. Since its arrival in China, FAW populations have quickly and extensively spread from the south to the north of China, causing serious damage to local agricultural commodities [40] and exhibiting absolutely competitive advantages over native lepidopteran pests [41].

Larvae of *S. frugiperda* are subjected to larval competition and consequent cannibalism [3]. A previous study has demonstrated that cannibalism of *S. frugiperda* larvae mainly occurs after reaching the fourth instar stage and is associated with higher larvae density [42]. Cannibalistic FAW larvae exhibited lower pupal weights and slower development [43], which might also be beneficial to the survival under food deprivation conditions and colonization of new habitats [3]. Evidence of larval density associated with population dynamics and behavioral traits of adults has been reported in most migratory species [10,44]. Increased larval density results in an aggregation of adults, which leads to population outbreaks within a short time in the field, similar to locusts [15]. Long-distance migration and high reproductive output are two key determinants of FAW evolutionary success and invasion status [45,46]. However, little is known about the effects of larval density on adult migration behavior of pests with outstanding cannibalism behavior such as the FAW. To better address the question above, in this study, we investigated the effects of larval density on pupal weight, body weight, wing expansion, and reproductive and flight performances of *S. frugiperda* and then characterized population dynamics. Our work may aid in providing baseline information for determining and controlling this pest.

## 2. Materials and Methods

### 2.1. Insect Rearing

The FAW larvae (corn strain) used in this study were collected from a maize field in Nanning, Guangxi Zhuang Autonomous Region, China (108.37° E, 22.82° N), in May 2019 and reared in the laboratory for one generation before the initiation of the experiments. The larvae were supplied with fresh corn seedlings until pupation in a glass jar (850 mL). Pupae were placed on sterilized soil with 15% moisture. After eclosion, newly emerged adults were individually paired in cylindrical plastic cages (1000 mL) and provided with 10% honey water. This colony was maintained at 27 ± 1 °C and 60% ± 5% relative humidity, with a 16-h light: 8-h dark photo period.

### 2.2. Larval Density Treatment

Newly hatched FAW larvae were reared at densities of 1, 5, 10, 20 and 30 larvae per 850 mL glass jar throughout the whole feeding period. Evidence of cannibalism associated with high larval density has been reported in *S. frugiperda* [3,42]. We examined the number of surviving larvae daily and replaced the missing larvae with larvae of the same size and density to maintain constant densities.

### 2.3. Pupal Weight, Body Weight and Wing Expansion

Pupal weight of 4-d-old pupae was measured using an electronic balance (Feihong Electronics Co., Ltd., Beijing, China). The sample sizes for pupal weight from 1 to 30 larvae per jar were 71, 37, 65, 41 and 60, respectively. After weighing, the pupae were put back into the original glass jar until eclosion. When newly emerged *S. frugiperda* adults from five larval density treatments were anesthetized with ether, the body weights and wing expansion (the length between the tip of the apical angle of the left and right forewing) of 1-d-old female and male adults after emergence were determined using electronic balance and electronic Vernier calliper (Hebei Baigong Industrial Co., Ltd., Beijing, China), respectively. Sample sizes for body weight (from low to high rearing density) were 50, 34, 27, 36 and 38 females, and 51, 33, 52, 38 and 47 males, respectively. Sample sizes for wing expansion (from low to high rearing density) were 44, 30, 27, 35 and 36 females, and 53, 27, 50, 37 and 38 males, respectively.

### 2.4. Reproductive Parameter Determination

To explore the effects of larval density on adult reproduction in *S. frugiperda*, reproductive parameters were measured using the method described in our previous studies [16,47,48]. All 1-d-old adults from the different density treatments were separately paired (one female and one male) and provided with 10% honey water in cylindrical plastic cages (1000 mL), as described above. The oviposition (including the number of eggs and duration of lying per female) and mortality were observed and recorded daily until all the adults died. These data were used to determine the lifetime fecundity, preoviposition period (POP), oviposition period, period of first oviposition (PFO) and adult longevity. The POP refers to the number of days between adult emergence and the first oviposition event. The oviposition period refers to the duration from the first to last oviposition event. The PFO was used to evaluate the synchronization of the first oviposition event, which was defined as the duration between one female’s POP and the minimal POP [47,49]. The sample sizes for each density treatment were 23, 30, 24, 28 and 20 pairs from low to high rearing density, respectively.

### 2.5. Flight Capacity Determination

Flight tests were conducted by using a 48-channel computerized flight mill system [16]. Individual moths on day 1 after emergence were attached to the end of the flight mill arm of a round-about flight mill (diameter = 20 cm), as described in previous studies [15,49]. A 10-h flight test (from 20:00 to 06:00), conducted under dark conditions at 25 ± 1 °C and 60% ± 5% RH, was initiated. The recordings of flight parameters (namely, the total flight duration, flight distance and average speed) were performed automatically by a computer. The number of replicates for each density treatment (from low to high) were 45, 29, 27, 36, 29 females, and 49, 28, 43, 32 and 36 males, respectively.

### 2.6. Data Analysis

All data obtained in this study are presented as means ± standard errors. The effects of larval density on pupal weight, body weight, wing expansion, and reproductive and flight parameters were assessed with one-way analysis of variance followed by Tukey’s honestly significant difference (HSD) post hoc test (*p* < 0.05). Pearson correlation analysis was conducted to examine the effects of body weight and wing expansion on reproductive and flight capacity in adults. All statistical analyses were performed using SPSS software (version 22.0; SPSS).

## 3. Results

### 3.1. Effect of Larval Density on Pupal Weight, Body Weight and Wing Expansion in Spodoptera frugiperda

The results showed that the pupal weight decreased significantly with increasing larval density (*F*_4,269_ = 16.18, *p* < 0.001). The weight of pupae reared as solitary larvae was significantly higher than that of pupae reared at densities of 10, 20 and 30 larvae per jar (Figure 1). We found that larval density had significant effects on the body weights of female and male adults (female: *F*_4,180_ = 24.23, *p* < 0.001; male: *F*_4,216_ = 36.85, *p* < 0.001). With increasing larval density, the body weights of adults decreased continuously. The body weights of female and male adults reared solitarily were significantly higher than those of adults reared at densities ranging from 5 to 30 larvae per jar. The female and male adults from the 30 larvae per jar treatment group had the lowest body weights, which were significantly lower than those in the other treatment groups (Figure 2A,C).

Furthermore, the degrees of wing expansion in female and male adults were significantly affected by larval density (female: *F*_4,167_ = 14.47, *p* < 0.001; male: *F*_4,200_ = 10.975, *p* < 0.001). The degree of wing expansion in females reared at a density of 30 larvae per jar was significantly lower than those in females reared at the other densities (Figure 2B). Similarly, male moths from the 30-larvae treatment group had the lowest degree of wing expansion, which was significantly less than those in the 1-, 10- and 20-larvae treatment groups (Figure 2D).

### 3.2. Effect of Larval Density on Reproduction in Spodoptera frugiperda

The reproductive performance of female FAWs was negatively correlated with larval density (Table 1). The POP of females reared solitarily was significantly reduced compared to that of females in the 10-larvae treatment group (*F*_4,120_ = 2.72, *p* = 0.033); however, it did not differ from those of females in the other density treatments (Table 2). Similarly, the PFOs of females in the solitary and 5-larvae treatment groups were also shorter than that of adults from the 10-larvae treatment group (*F*_4,120_ = 4.04, *p* = 0.004, Table 2). The lifetime fecundity of female adults decreased significantly with increasing larval density from 1494.00 to 750.65 (*F*_4,120_ = 10.31, *p* < 0.001). Females from the solitary larva treatment group produced more eggs than adults from the other four larval treatment groups (Table 2). However, larval density did not significantly affect the oviposition period (*F*_4,120_ = 1.68, *p* = 0.160) or adult longevity (female: *F*_4,120_ = 1.73, *p* = 0.147; male: *F*_4,120_ = 0.97, *p* = 0.429, Table 2). Furthermore, body weight and wing expansion had positive correlations with reproduction in female FAWs (Table 1). In conclusion, female adult FAWs reared as solitary larvae had a stronger reproductive performance than those reared under larval density conditions.

### 3.3. Effect of Larval Density on Flight Capacity in Spodoptera frugiperda

Larval density was negatively correlated with flight capacity in female and male FAWs (Table 3). In females, the flight duration of female moths reared solitarily was the longest and was significantly longer than those of moths reared at densities of 20 and 30 larvae per jar (*F*_4,161_ = 3.63, *p* = 0.007, Figure 3A). Similarly, females reared solitarily had a longer flight distance (*F*_4,161_ = 10.71, *p* < 0.001, Figure 3B) and flight velocity (*F*_4,161_ = 10.52, *p* < 0.001, Figure 3C) than those reared as densities of 5, 20 and 30 larvae per jar.

In males, adults reared solitarily showed significantly longer flight durations than those reared at other densities (*F*_4,183_ = 5.642, *p* < 0.001, Figure 3B). The flight distance of males decreased significantly with increasing larval density (*F*_4,183_ = 5.01, *p* = 0.001). The flight distances of males reared at densities of 1, 5 and 10 larvae per jar were significantly longer than that of males reared at a density of 30 larvae per jar (Figure 3D). The flight velocity of males reared at a density of 30 larvae per jar was also significantly lower than those of males reared at densities of 5 and 10 larvae per jar, but it was not different from those of males reared at the other densities (*F*_4,183_ = 4.76, *p* = 0.001, Figure 3F).

The flight capacity of female and male FAWs was also positively associated with body weight and wing expansion (Table 3). Pupal weight had no significant correlation with flight capacity in females but showed a positive correlation with flight distance in male FAWs (Table 3). Thus, FAWs reared solitarily had a better flight capacity than those reared at higher larval densities.

## 4. Discussion

Increasing evidence has indicated that larval density is associated with changes in flight and reproduction [4,10,15,44]. However, research on density-mediated flight and reproduction in the agricultural pest *S. frugiperda* is lacking. Our study showed that alterations in flight and reproduction of *S. frugiperda* adults were associated with different larval densities, which is consistent with previous reports in *M**. separata* [4,16], *L**. sticticalis* [10,47], *Cnaphalocrocis medinalis* [48,49] and *Locusta migratoria* [3,44,49]. We found that *S. frugiperda* reared solitarily exhibited higher pupal and body weights and stronger reproductive and flight capacities than those reared at higher larval densities.

In our study, the pupal weight and body weight of *S. frugiperda* decreased significantly with increasing larval density, consistent with prior reports that larval crowding negatively influenced the pupal weight and body weight in *M. separata* [4], *L. sticticalis* [10], *Athetis lepigone* (Moschler) [50] and other insect species. Immature FAW exhibits high levels of larval competition and cannibalism [43,51]. The levels of larval competition and cannibalism rise with increasing larval crowding [3], and these behaviors are aggravated by the aggregated distribution of larvae, similar to granary weevils (*Sitophilus granarius* L.) [52]. Previous studies indicated that cannibal FAWs had a lower pupal weight than larvae provided with corn leaves [3,43]. We also found that the wing expansion in *S. frugiperda* adults differed significantly under different larval-density conditions. The degree of wing expansion in FAW adults reared solitarily was significantly greater than that in those reared at a density of 30 larvae per jar. However, no differences were observed in *L. sticticalis* [53] and *L. migratoria* [3]. These results were consistent with a previous study in which the increase in larval density decreased pupal weight and forewing width in FAW from the laboratory population [54]. Pearson correlation analysis also demonstrated that body weight and wing expansion were positively associated with flight and reproductive capacity in *S. frugiperda.*

Environment-dependent polyphenism of reproduction and flight are major adaptive strategies that strengthen the ecological success of many migratory insects [15,16,55,56,57]. Indeed, evidence of larval population density associated with phase variations between the morphs of migrants and residents has been reported in wing-monomorphic insects, such as the *M. separata* [17,18], *S*. *exigua* [29] and *L**. sticticalis* [11]. Migrants produced as a result of larval crowding exhibited a stronger flight performance than residents reared solitarily as larvae [58]. In *L. sticticalis* and *M. separata*, moderate larval crowding triggers migratory behavior, but overcrowding decreases the migration propensity because physiological stress reduces flight capacity [4,10]. However, migration propensity in *C. medinalis* adults is not enhanced under crowding conditions during the larval phase [59]. In *H**. armigera*, high larval density also does not increase their flight capacity [60]. Our results showed that crowded larval population densities decreased the flight performance of *S. frugiperda* in both sexes, resulting in solitary FAWs having a better flight capacity than those reared at higher larval densities. This result was consistent with some previous reports that high larval density did not produce pre-migrant traits, as larval density had little effect on developmental time, adult activity, whole-body weight and lipid contents in FAW [61,62]. Our findings might be attributed to high levels of cannibalism among *S. frugiperda* larvae. Previous studies found that the distribution pattern of *S. frugiperda* larvae was closely associated with cannibalism [63,64,65]. Larvae of *S. frugiperda* between the first and third instar showed a pattern of aggregation distribution, but the later larvae exhibited an even distribution in maize and wheat fields [63,64]. He et al. [3] reported that cannibalism could occur when as few as two fourth-instar larvae were provided with sufficient and nutrient-rich food in *S. frugiperda*. Wang et al. [42] found that cannibalism of *S. frugiperda* larvae mainly occurred at the fifth instar stage. Cannibalism can reduce competition and obtain additional high-value nutritional resources while decreasing search time [25,66], which may aid *S. frugiperda* survival under starvation and successfully colonize new food plants [3]. We thus deduced that cannibalism might reduce competition among *S. frugiperda* individuals reared under larval crowding conditions, resulting in a decreased flight potential and flight capacity of subsequent adults.

Crowded conditions reportedly decrease the reproductive performance of many insects, such as *Mythimna convecta* [67], *M. separata* [4,58], *S. exigua* [29], *L. sticticalis* [10] and *C. medinalis* [59]. Our result was consistent with these observations, as increasing the intensity of larval crowding significantly restrained reproduction in *S. frugiperda* adults, resulting in a delay of first oviposition and decreases in synchronized oviposition and lifetime fecundity. These results indicate that FAWs reared solitarily as larvae exhibit a stronger reproductive performance. Furthermore, a negative correlation between larval density and reproduction also existed in *S. frugiperda*. Thus, larval density was one of the important factors affecting adult migration in many migratory insects without cannibalism or with low levels of cannibalism. Whereas, for those insects with high levels of cannibalism, larval crowding aggravated internal competition and cannibalism, which resulted in increased mortality, delayed development time and decreased adult reproductive and flight capacity. We speculated that larval crowding would not be the main factor inducing adult migration in FAW, owing to no significant enhancement of adult migration propensity and capacity for adult migration. Therefore, more attention should be paid to the effects of temperature, wind, photoperiod and other weather factors on reproduction and migration to better predict outbreaks in FAW.

## 5. Conclusions

In conclusion, our results suggest that alterations in flight and reproductive traits may be strongly dependent on the level of larval crowding in *S. frugiperda*, which may also be a widespread adaptive strategy in many migratory insects to cope with environmental stressors during the larval phase. FAWs reared solitarily exhibited higher pupal and body weights and stronger reproductive and flight capacities than those reared at higher larval densities. Larval crowding did not produce pre-migrant traits and enhance the migration propensity in *S. frugiperda* adults. Therefore, larval crowding may not be a key factor in inducing the migratory behavior in *S. frugiperda* in the case of high levels of cannibalism. These findings may enhance our understanding of the adaptive behavioral strategies of *S. frugiperda* in response to unfavorable conditions and provide novel strategies for the control of this invasive pest.

## Figures and Tables

**Figure 1 insects-13-00581-f001:**
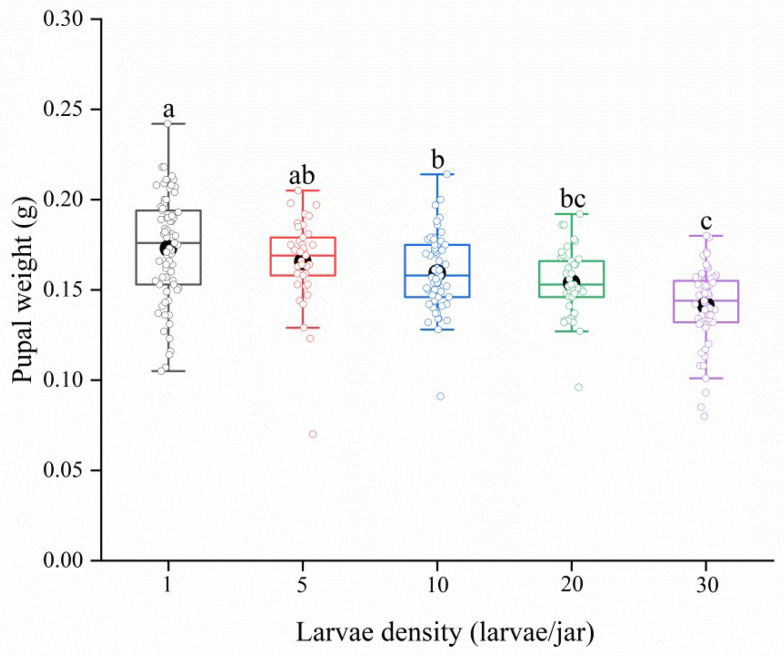
Pupal weight of *Spodoptera frugiperda* under different larval densities. Solid black bars represent the maximum line, median line and minimum line (from top to bottom in each boxplot). Boxplots display a mean dot, IQR boxes, 1.5*IQR whiskers and observed values (open circle). Different lowercase letters above group bars indicate significant differences among different larvae densities by Tukey’s HSD test at 5% level. The sample sizes from left to right are 71, 37, 65, 41 and 60, respectively.

**Figure 2 insects-13-00581-f002:**
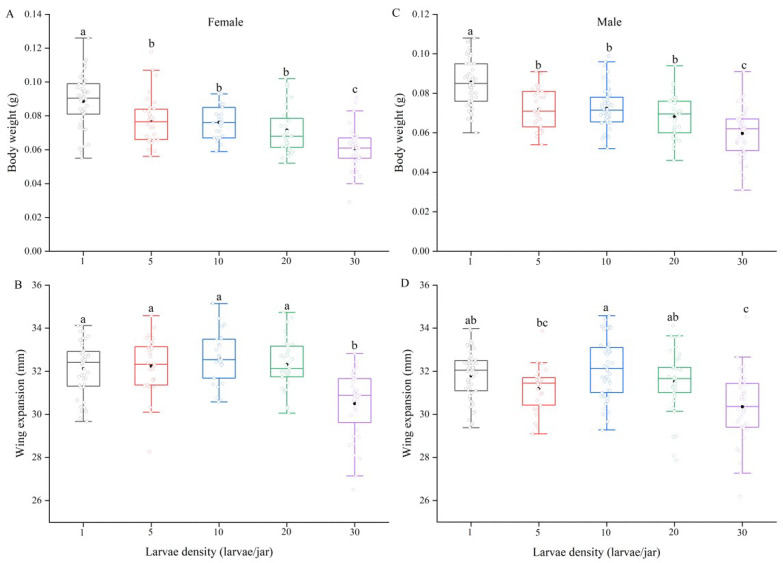
Body weight (**A**: females, **C**: males) and wing expansion (**B**: females, **D**: males) of *Spodoptera frugiperda* under different larval densities. Solid black bars represent the maximum line, median line and minimum line (from top to bottom in each boxplot). Boxplots display a mean dot, IQR boxes, 1.5*IQR whiskers and observed values (open circle). Different lowercase letters above group bars indicate significant differences among different larvae densities by Tukey’s HSD test at the 5% level. Sample sizes for body weight (left to right) were 50, 34, 27, 36 and 38 females, and 51, 33, 52, 38 and 47 males, respectively. Sample sizes for wing expansion were 44, 30, 27, 35 and 36 females, and 53, 27, 50, 37 and 38 males, respectively.

**Figure 3 insects-13-00581-f003:**
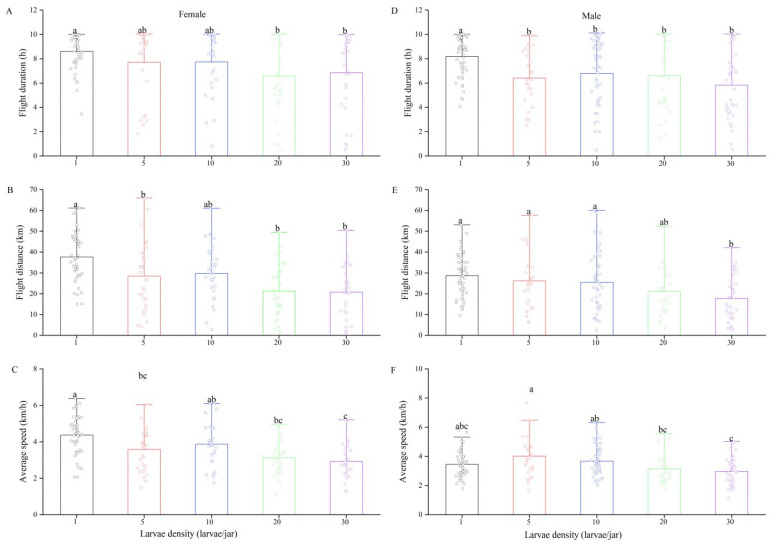
Flight duration (**A**: females, **D**: males), flight distance (**B**: females, **E**: males) and flight velocity (**C**: females, **F**: males) of *Spodoptera frugiperda* adults under different larval densities. Different letters above group bars indicate significant differences among different larvae densities (1-way ANOVA: *p* < 0.05). Sample sizes for female moths were 45, 29, 27, 36, 29 from left to right; the sample sizes of male worms were 49, 28, 43, 32 and 36.

**Table 1 insects-13-00581-t001:** Pearson correlations of reproduction of *Spodoptera frugiperda* under different larval densities, with body weight and wing expansion.

Parameters	Larval Densities		Body Weight		Wing Expansion	
	*r*	*p*-Value	*r*	*p*-Value	*r*	*p*-Value
Preoviposition period	0.028	0.642	0.448	<0.001 *	0.532	<0.001 *
Period of first oviposition	0.091	0.133	0.307	<0.001 *	0.328	<0.001 *
Lifetime fecundity	−0.171	0.005 *	0.482	<0.001 *	0.604	<0.001 *
Oviposition period	−0.052	0.390	0.484	<0.001 *	0.632	<0.001 *
Female longevity	−0.023	0.708	0.488	<0.001 *	0.605	<0.001 *
Male longevity	0.008	0.893	0.499	<0.001 *	0.633	<0.001 *

* Significant *p*-values.

**Table 2 insects-13-00581-t002:** Reproductive performances of *Spodoptera frugiperda* adults under different larval densities.

Larvae Density (Larvae/Jar)	Pre-Oviposition Period (d)	Period of First Oviposition (d)	Lifetime Fecundity (Per Female)	Oviposition Period (d)	Female Longevity (d)	Male Longevity (d)
1	4.74 ± 0.30 b	1.74 ± 0.30 b	1494.00 ± 157.27 a	5.39 ± 0.49 a	11.48 ± 0.66 a	12.48 ± 0.94 a
5	5.90 ± 0.34 ab	1.90 ± 0.34 b	1104.47 ± 44.05 b	5.50 ± 0.22 a	13.63 ± 0.52 a	13.67 ± 0.55 a
10	7.04 ± 0.82 a	4.04 ± 0.82 a	926.96 ± 80.30 b	5.00 ± 0.32 a	14.08 ± 1.17 a	13.71 ± 0.97 a
20	6.75 ± 0.52 ab	3.75 ± 0.52 ab	812.61 ± 63.93 b	4.39 ± 0.32 a	12.86 ± 0.55 a	14.64 ± 0.65 a
30	6.20 ± 0.57 ab	3.20 ± 0.57 ab	750.65 ± 91.03 b	5.25 ± 0.48 a	12.50 ± 0.84 a	14.10 ± 1.01 a

Data are shown as mean ± SEM. Different lowercase letters in the same column indicated significant differences among different larvae densities by Tukey’s HSD test at the 5% level. The sample sizes from top to bottom were 23, 30, 24, 28 and 20.

**Table 3 insects-13-00581-t003:** Pearson correlations of flight distance, flight velocity, and flight duration of *Spodoptera frugiperda* adults under different larval densities, with pupa weight, body weight and wing expansion.

Sex	Parameters	Flight Duration		Flight Distance		Flight Velocity	
		*r*	*p*-Value	*r*	*p*-Value	*r*	*p*-Value
Female	Larval densities	−0.147	0.015 *	−0.246	<0.001 *	−0.200	0.001 *
	Pupa weight	−0.046	0.447	−0.012	0.842	0.001	0.993
	Body weight	0.714	<0.001 *	0.668	<0.001 *	0.755	<0.001 *
	Wing expansion	0.750	<0.001 *	0.692	<0.001 *	0.800	<0.001 *
Male	Larval densities	−0.168	0.005 *	−0.215	<0.001 *	−0.145	0.016 *
	Pupa weight	0.067	0.269	0.122	0.043 *	0.109	0.072
	Body weight	0.602	<0.001 *	0.543	<0.001 *	0.596	<0.001 *
	Wing expansion	0.697	<0.001 *	0.610	<0.001 *	0.723	<0.001 *

* Significant *p*-values.

## Data Availability

The data presented in this study are available on request from the corresponding author.

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
