# Peer review of "Larval Crowding Did Not Enhance Adult Migration Propensity in Spodoptera frugiperda"

_insects, 2022, doi:10.3390/insects13070581_

Round 1

Reviewer 1 Report

Overall I found the manuscript to be well written, concise, and overall sound. The work adds to the literature regarding the effects of larval crowding on adult flight and reproductive traits - with adults resulting from increasingly crowded larval rearing treatments displaying poorer reproductive and flight capacities/performance. The research design and analyses are appropriate and the discussion well crafted and supported by the literature. beyond some minor edits identified in the attachment, my only suggestion is to produce additional clarity about treatment replication. It is unclear how many replications were undertaken. If limited replication was performed, I would suggest providing a clear reason for this and an explanation about the resulting limitations of the study.

Line 65: Clarify statement “make some negative influence.”

Line 91: extra period at the end of the sentence.

Line 109: S. frugiperda needs to be in italics

In the Methods section under Larval Density Treatment- please provide details regarding the number of replications performed

Line 240: Please clarify this sentence. It appears vague. Should this read “Increasing evidence has indicated that larval density is associated with changes in flight and reproduction”? Seems like this should read more like line 312.

Author Response

Dear Reviewer:

Thanks very much for reviewers' suggestions on our manuscript entitled “Larval crowding did not enhance adult migration propensity in Spodoptera frugiperda” (manuscript # insects-1783322). We have read these comments carefully and made major modification correspondingly. We have labelled all the modifications we have made in pink and also the answers to these questions including line numbers. In addition, changes in the revised manuscript were highlighted using red font.

Once again, we acknowledge your comments very much, which are valuable and meaningful in improving the quality of our manuscript.

Sincerely yours.

Weixiang Lü

2022-6-23

Reviewer 2 Report

Dear authors,

I have reviewed your manuscript and find it interesting to read. Below are some suggestions for improvement

1.       The direct implication of S. frugiperda larval crowding is on level of cannibalism as acknowledged in your introduction and many parts in the discussion. Unfortunately, no data was provided on cannibalism. This data should be there since you kept replacing dead individuals. Why was the data not presented and yet a lot of reference is made to cannibalism?

2.       In our unpublished lab results, we observed that larvae that ate conspecifics (FAW) were bigger and more aggressive than those fed on maize leaves alone

3.       It would be interesting to see the total number of larvae used in each treatment

4.       The title is. 1) restrictive to migration and not covering the whole work; 2) not consistent with the data in figure 3

5.       Be consistent in the use of Spodoptera frugiperda, fall armyworm in FAW in section titles, table titles and figure legends

6.       It has to be clearly stated what the utility of this research results is in as far as managing FAW is concerned

Simple summary

Line 22, replace “stress condition” with “density”

Abstract line 35, move “not” to come immediately after “may”; delete “in” before “affecting”

Introduction, line 77, delete the second “l” in the word absulutel

Methods

How many individuals were used per treatment and how many replications were used

If cannibalism is a known behavior of S. frugiperda, why did you go ahead to replace dead larvae instead of simply taking data on cannibalism and the stage at which cannibalism is highest?

Line 148, it is…honesty significant …….

Results

The order of presentation of results in text should be consistent with the order of presentation of tables and figures

Table titles, replace “from” with ”under”

Table 2 – transfer significant test interpretation to footnote (line 213)

Discussion

Line 244, replace “was” with “is”

Lines 244 to 246, abbreviate genus names is stated earlier

Line 267, strengthen

Line 270, scientific names

Conclusion

Lines 318 to 318, why is the conclusion in contrast to the observation of superior performance when a single larva was reared in a container?

Author Response

(The authors gave the same response as above.)

Reviewer 3 Report

The manuscript ID: insects-1783322 "Larval crowding did not enhance adult migration propensity in Spodoptera frugiperda" makes a good impression. Authors investigated the effects of larval density on flight and reproductive parameters of S. frugiperda adults. Of course, this is an important and relevant topic, since this dangerous pest is the subject of attention for many countries of the world. The researchers used adequate methods for obtaining and analyzing the results. Three clear and meaningful tables and three figures accompany this article, which contributes to a clear structuring of the material presented. The conclusions are consistent with the evidence and arguments and they address the main question posed.

I believe that the article will be of interest to a wide range of readers and can be published in Insects after some corrections.

Line 75 [38-9] – I don't understand what references should be. May be is it [38-39]?

Line 107 Please write the number of repetitions for each density treatment.

Lines 161-163. I'm not sure the sentence conveys the meaning correctly. “than those in the higher-density treatment groups” – “than those in the other treatment groups”, not higher.

Typo: Ovipostion period (d) (Table 1).

In table 3, all parameters except “pupa weight” are capitalized.

Author Response

(The authors gave the same response as above.)
